# Locating the Coseismic Ionospheric Disturbance from the 2010 Mw7.8 Mentawai, Indonesia, Tsunami-Earthquake

Cedric Twardzik 1, Lucie M. Rolland 1, Edhah Munaibari 1, and Thomas Dylan Mikesell 2

Correspondence: Cedric Twardzik (twardzik@geoazur.unice.fr)

#### Abstract.

Forecasting tsunamis using seismological and geodetic observations remains challenging. However, tsunamis generate acoustic waves that travel into the atmosphere and affect the ionosphere, creating disturbances in total electron content (TEC). These disturbances, known as Coseismic Ionospheric Disturbances (CIDs), can be observed using GNSS data. Identifying the source location of the CID could provide valuable information about whether a tsunami has been triggered. In this study, we attempt to determine the source location of the CID following the 2010 M<sub>w</sub>7.8 Mentawai tsunami-earthquake, using onset time measurements of the CID on vTEC waveforms. Our approach combines acoustic wave propagation within a realistic atmosphere alongside a Bayesian inversion framework that considers uncertainties in the observations and the forward problem itself. Our approach search for the effective height of the ionosphere rather than fixing its altitude, as is commonly done. Our results suggest an effective height of the ionospheric of around 250 km, which is below the F-layer peak. Furthermore, we show that the inferred source location of the CID does not coincide with the zone of maximum seafloor uplift or the area where the tsunami initiates. An alternative model based on the assumption of a homogeneous atmosphere locates the source of the CID within the area where the tsunami initiates. While this might imply that the realistic atmosphere model does not allow acoustic waves to propagate at the correct speed, our findings demonstrate that both models have the same apparent acoustic wave speed. Therefore, we argue that the approach using a realistic atmosphere provides an accurate location of the CID source because it takes the complexity of propagation paths into account. We offer several explanations as to why the source location the CID does not match the area where the tsunami initiates, such as the effect of the spatial extent of the actual source, more complex tsunami initiation dynamics due to bathymetry effects, or potential errors in the earthquake source models that could shift the area where the tsunami initiates. This work demonstrates the potential of using ionospheric observations to complement existing tsunami monitoring strategies.

# 1 Introduction

Large subduction earthquakes pose a major threat to nearby populations because of the ground shaking, but also because of their ability to generate tsunamis. Tsunamis often reach coastal areas within several tens of minutes after an earthquake, providing a potential window for people to reach safety (e.g., Tsushima and Ohta, 2014). However, to issue an effective warning, it

<sup>&</sup>lt;sup>1</sup>Université Côte d'Azur, Observatoire de la Côte d'Azur, CNRS, IRD, Géoazur, Valbonne, France

<sup>&</sup>lt;sup>2</sup>Norwegian Geotechnical Institute, Natural Hazards, Oslo, Norway

50

is crucial to know whether a tsunami has actually been generated. Once the earthquake's magnitude has been determined, a warning can be issued if the earthquake is considered large enough to generate a tsunami. Unfortunately, it is only possible to observe that one is actually on its way when it reaches the first coastal tide gauges or deep-ocean pressure gauge. Therefore, observations close to the source could provide more warning time. Although developing seafloor instruments could help (e.g., Becerril et al., 2024), this remains a challenge.

Because of the coupling between the ocean and the atmosphere, a tsunami generates acoustic waves that propagate upwards into the atmosphere. They eventually reach the ionosphere, causing a perturbation in the electron density (Figure 1). The first observations of these disturbances were made using radio signals (e.g., Davies and Baker, 1965; Yuen et al., 1969). Nowadays, it can be done by analyzing data from ground-based Global Navigation Satellite System (GNSS) receivers. This is because these ionospheric disturbances affect the propagation of the signals between satellites and receivers (e.g. Artru et al., 2005; Astafyeva, 2019). Although these are not direct observations of the tsunami, they can still be useful for an early warning system, since the tsunami's signature in the ionosphere can be detected as early as 10 to 15 minutes after an earthquake, well before the tsunami reaches tide gauges (Kherani et al., 2016). Therefore, monitoring the electron density, or Total Electron Content (TEC), using GNSS data after an earthquake may provide an reliable indication of whether a tsunami has been generated, as first suggested by Najita et al. (1974). While the detection of TEC anomalies remains the first step of a potential warning system, and some progress having been made in that direction in recent years (e.g., Manta et al., 2020; Brissaud and Astafyeva, 2024; Maletckii and Astafyeva, 2024; Luhrmann et al., 2025), this study focuses on locating the source of the co-seismic ionospheric disturbances (CID) in order to determine whether they originate onshore or offshore.

One approach for locating the source of CID is to model the full TEC waveform (e.g., Zedek et al., 2021; Inchin et al., 2024). However, this approach relies on intensive numerical simulations and may not be suitable for real-time or near-real-time warning systems, although promising results having been obtained in this direction (e.g., Sanchez et al., 2025). An alternative approach, more observation-oriented, uses the coherence between the TEC waveforms obtained from an array of GNSS receivers to perform back-projection (e.g. Gómez et al., 2015; Lee et al., 2018). A disadvantage of this approach is that it requires an array of GNSS receivers, which may not always be available. Therefore, the most widely used approach is to locate the source of CID using onset time measurements from TEC waveforms.

Numerous studies have explored how to locate the source of CID from onset time measurements (e.g. Liu et al., 2006, 2010; Tsai et al., 2011; Astafyeva et al., 2013; Srivastava et al., 2021). However, many of these studies use assumptions that may introduce biases in source location. For example, it is commonly assumed that CID propagation follows a two-stage path (e.g. Liu et al., 2006, 2010). First, the acoustic wave travels vertically to a given height (typically 300 km) before continuing horizontally to the sub-ionospheric piercing point (SIPP), where the acoustic wave front intersects the line of sight of a particular satellite-station pair. Another common assumption is that the acoustic wave travels through a homogeneous atmosphere before reaching the SIPP (e.g. Kiryushkin and Afraimovich, 2007). Once again, the ionospheric height is set at a specific altitude. Consequently, most analyses disregard the complex structure of the atmosphere when modelling the propagation of the acoustic waves. Furthermore, the effective height of the ionosphere is often fixed. The aim of this study is to evaluate the impact of removing these assumptions. To do so, we focus on the 2010 M<sub>w</sub>7.8 Mentawai Islands, Indonesia, tsunami earthquake (Figure

2), which is known to have generated a tsunami with detectable CID on the TEC waveforms (Manta et al., 2020; Heki et al., 2022).

Tsunami-earthquakes generate significantly larger tsunamis than would be expected based on their surface wave magnitude (e.g. Kanamori and Kikuchi, 1993). Such earthquakes are often characterised by an unusually long source duration compared to that expected from scaling laws (e.g. Tsuboi, 2000). For example, on 25 October 2010, a M<sub>w</sub>7.8 earthquake occurred off the coast of the Mentawai Islands and generated a large tsunami, resulting in 450 fatalities, as reported by the International Tsunami Information Centre. The expected source duration for a M<sub>w</sub>7.8 earthquake is about 45 seconds, according to the scaling relation from Duputel et al. (2012). Instead, Lay et al. (2011) determined that the source duration of that earthquake was closer to 90 seconds. This leads to an unusually low amount of radiated energy for the given seismic moment, making it more difficult to assess the real tsunami threat using conventional seismological data-based methods (Lay et al., 2011). As discussed above, studying CID could provide an effective means of monitoring the potential for significant tsunamis, even in the case of particular events.

This study investigates the source location CID generated by the  $2010 \, M_w 7.8$  Mentawai tsunami-earthquake based on onset time measurements from TEC waveform. Specifically, the study examines the impact of considering the complex propagation of acoustic waves through the atmosphere, as well as the effect of relaxing the common assumption that the effective height of the ionosphere is fixed. A Bayesian framework is employed to evaluate the model parameters and their associated uncertainties. The next section provides a more detailed description of the data and methods used. This is followed by a description of the results, which are then discussed.

### 2 Data and Methods

## 2.1 Onset time measurements of the CID on TEC waveforms

The TEC waveforms are derived from the data recorded by 28 GNSS receivers from the Sumatran GPS Array (SuGAr), with each receiver locking onto between 4 and 11 satellites (Figure 2). Each satellite transmits signals to a given receiver at two different frequencies,  $f_1$  and  $f_2$ , which are 1575.42 MHz and 1227.60 MHz, respectively. Due to the dispersive nature of the ionosphere, signals at theses frequencies are affected differently as they passing through the ionospheric layer. This property makes it possible to obtain the so-called slant TEC from the raw data:

85 sTEC = 
$$\frac{1}{40.3} \left[ \frac{f_1^2 f_2^2}{f_1^2 - f_2^2} \right] [(L_1 - L_2) - (d_r + d_s)]$$
 (1)

where  $L_1$  and  $L_2$  are the carrier phases for the two frequencies  $f_1$  and  $f_2$ , respectively, and  $d_r$  and  $d_s$  are the code delay biases for the receiver and satellite, respectively. To normalize the TEC amplitude and to avoid the effects of low elevation angles, the slant TEC is converted to vertical TEC (Astafyeva, 2019):

$$vTEC = sTEC \cdot \cos \left[ \arcsin \left( \frac{R_z}{R_z + H_{ion}} \cos(\theta) \right) \right]$$
 (2)

100

where  $R_z$  is the radius of the Earth,  $H_{ion}$  is the assumed height of the ionospheric thin shell (300 km), and  $\theta$  is the elevation angle of the satellite. There are 217 vTEC time series that record vTEC variations for at least 3600 seconds after the earthquake's origin time given by the Global Centroid Moment Tensor (GCMT) solution (2010-10-25 14:42:22.50 UTC). The short-term vTEC variations can be enhanced by removing the long-period variations due to diurnal ionospheric variability, satellite motion and quasi-constant instrumental biases. This is done by filtering the vTEC waveforms using an one-pass third-order Butterworth bandpass filter (Rolland et al., 2013) with a frequency range of 2–10 mHz, which is consistent with Manta et al. (2020).

The CID onset time is measured manually by inspecting simultaneously the vTEC waveforms together with their first and second time derivatives, and without visual cues from pre-computed theoretical arrival times. A total of 51 onset time measurements are made from the 217 vTEC waveforms (Figure 3 for an example).

These onset time measurements are refined using the iterative cross-correlation and stacking (ICCS) algorithm (Lou et al., 2013). The algorithm aims to determine the optimal time shift to apply to the measured onset times so that the vTEC waveforms stack as coherently as possible. The alignment procedure is done on the first-time derivatives of the vTEC waveforms, with a time window of 375 seconds, comprising 125 seconds before and 250 seconds after the onset time. As the signature of CID depends heavily on the satellite, the ICCS algorithm is applied to each satellite independently. During the procedure, when a given vTEC waveform does not correlate with the stack at a correlation coefficient greater than 0.25, the measured onset time remains at its original value. Figure 4 illustrates the updates made to the measured onset times after applying the ICCS algorithm.

The ICCS algorithm is also used to quantify observations uncertainties  $(\sigma)$ . This is done by comparing the measured onset times  $(t_i)$  with the final onset times after applying the ICCS algorithm  $(t_f)$ . For a given observation,  $\sigma$  is set as  $\sqrt{(t_i-t_f)^2}$ . For observations that have not undergone any time adjustments by the ICCS algorithm,  $\sigma$  is set to the lowest non-zero adjustment. For vTEC waveforms that do not correlate well with the final stack,  $\sigma$  is set to 330 seconds. This reflects the discrepancy observed between the measured onset times and those presented by Manta et al. (2020) for GNSS receivers BTHL and PTLO and using satellite PRN29. These estimates of observations uncertainties are used to build the data covariance matrix,  $C_d$ , which is a diagonal matrix with dimensions  $N \times N$ , where N is the number of onset time measurements. Figure 5 shows the dataset of CID onset times used in this study, along with the observations uncertainties. Numerical values are provided in Table 1 to 7. The observations uncertainties obtained using this approach are similar to those that could be estimated using an AI-based approach as illustrated by (e.g., Brissaud and Astafyeva, 2024) for other events. This suggests that the data covariance matrix is designed reasonably well.

## 2.2 Computation of travel times in a realistic atmosphere

The acoustic wave speed profile of the atmosphere at the location and time of the 2010 Mentawai earthquake is obtained from the NRLMSISE-00 model (Picone et al., 2002), which provides information on the atmosphere's chemical composition and temperature as a function of altitude. Therefore, all derived quantities vary with altitude. For brevity, this dependency on altitude is not explicitly written hereafter.

First, using the ideal gas law, pressure can be calculated using the following formula:

$$125 \quad P = \sum_{i} n_i \frac{R}{N_A} T \tag{3}$$

where  $n_i$  represents the number of particles per cubic meter, R represents the ideal gas constant,  $N_A$  represents the Avogadro's number and T represents the temperature in Kelvin. Here, we consider the following particles: He, H, N, O,  $N_2$ ,  $O_2$  and Ar. Then, the density can be calculated using the following formula:

$$\rho = \sum_{i} n_i \frac{m_i}{N_A} \tag{4}$$

where  $m_i$  represents the molar mass of each particle. Finally, the heat capacity at constant pressure  $(C_p)$  and the heat capacity at constant volume  $(C_v)$  are calculated using the following formulas:

$$C_{p} = \sum_{i} x_{i} \frac{5}{2} R + \sum_{j} x_{j} \frac{7}{2} R$$

$$C_{v} = \sum_{i} x_{i} \frac{3}{2} R + \sum_{j} x_{j} \frac{5}{2} R$$
(5)

where  $x_i$  are the mass fractions of monoatomic particles, and  $x_j$  are the mass fractions of diatomic particles. The acoustic wave speed is then obtained by combining these equations:

$$135 \quad c = \sqrt{\frac{C_p}{C_v} \frac{P}{\rho}} \tag{6}$$

Additional factors are also taken into account. Firstly, the dependence of the acoustic wave speed on azimuth due to horizontal winds is considered. This is determined from the 2014 horizontal wind model (Drob et al., 2015):

$$c'(\theta) = c + \cos(\theta)w_m + \sin(\theta)w_z \tag{7}$$

where  $\theta$  is the azimuth of the observation and  $w_m$  and  $w_z$  are the meridional and zonal winds, respectively. Secondly, Earth flattening transformation is applied to the profiles in order to account for Earth's curvature (Hill, 1972):

$$c_f(\theta) = c'(\theta) \frac{a}{a+h} \tag{8}$$

where a is the Earth's radius taken as 6378.137 km and h is the altitude. This leads to a decrease of  $\sim$ 4.5% in the acoustic wave speed at an altitude of 300 km, which is the typical height at which the ionosphere experiences maximum ionisation. Figure 6 shows the various components of the acoustic wave speed profile used to calculate azimuth-dependent travel times.

A grid of theoretical travel times is built using the eikonal solver described in Noble et al. (2014) and implemented as a Python package in Luu (2022). This method of calculating theoretical arrival times is similar to that described in Garcés et al. (1998). The grid extends from 0 to 1000 km in altitude, from 0 to 2000 km in source-receiver distance, and from 0 to 360° in azimuth (Figure 7). The calculation of travel times does not consider the effect of satellite motion during the propagation of the

180

acoustic wave. This is because this study only considers onset times of CID, meaning that the line of sight from the receiver to the satellite can be assumed to be fixed at the time of observation.

Three parameters are required to predict the arrival time for a given station-satellite pair: the effective height of the SIPP ( $h_{\rm iono}$ ), the distance between the source and the surface projection of the SIPP ( $\Delta_{\rm SIP}$ ), and the source-receiver azimuth ( $\theta_{\rm SIP}$ ). The last two are derived from the latitude ( $\lambda_{\rm src}$ ) and longitude ( $\phi_{\rm src}$ ) of the source and the surface projection of the SIPP. See Prol and Camargo (2015) for details about calculating the surface projection of the SIPP. The model parameters for the inversion are therefore as follows: [ $h_{\rm iono}$ ,  $\lambda_{\rm src}$ ,  $\phi_{\rm src}$ ]. An additional model parameter in the form of a constant (i.e., non-station-specific) correction term ( $\tau$ ) is added to all predicted arrival times. This term aims to correct for any systematic biases in acoustic wave propagation that have not been accounted for. However, it is not intended to account for potential errors in the NRLMSISE-00 model, which is addressed by introducing the prediction covariance matrix ( $C_p$ ).

Indeed, Emmert et al. (2021) shows that the NRLMSISE-00 model differs from in-situ observations in terms of temperature, particularly at altitudes lower than 100 km. The data presented in Emmert et al. (2021) lead to an average temperature difference of -1.4 K  $\pm$  6.8 K across various types of measurement. Temperature is known to be one of the most influential factors affecting acoustic wave speed. Additionally, David et al. (2015) suggests that the 2014 horizontal wind model has an uncertainty of about  $\pm$ 37 m/s. The introduction of  $C_p$  is intended to account for these known sources of error when calculating the arrival times of CID.

The prediction covariance matrix, which converts uncertainty in temperature and wind speed into uncertainty in predicted arrival times, is obtained empirically using the approach outlined in Caballero et al. (2023). To do this, predictions made using a reference model, a reference temperature profile and a reference wind speed profile are compared with a set of predictions made using the same reference model, but with temperature and wind speed profiles perturbed around their documented uncertainties:

170 
$$C_p = \frac{1}{n-1} \sum_{i=1}^{n} (y(\Psi^i, \mathbf{m}) - y(\Psi^0, \mathbf{m})) (y(\Psi^i, \mathbf{m}) - y(\Psi^0, \mathbf{m}))^T$$
 (9)

where  $y(\Psi^0, \mathbf{m})$  represents the predictions made using the reference model, the reference temperature profile and the reference wind speed profile, while  $y(\Psi^i, \mathbf{m})$  represents the the predictions made using the reference model but with a given perturbed temperature and wind speed profiles. For the reference model, the earthquake epicentre is used as the source location (3.49°S,  $100.08^{\circ}E$ ), the effective height of the ionosphere is set to 250 km and no uniform time shift is applied to the predictions. For  $y(\Psi^i, \mathbf{m})$ , a total of 2500 perturbed temperature and wind speed profiles are used, each drawn within the level of uncertainty described above.

Another factor that makes predicted arrival times uncertain is the assumption that the line of sight (LOS) is fixed at the onset time for each satellite-receiver pair. Since the onset times are uncertain, so too is the LOS. The same approach described above is used to calculate the prediction covariance matrix with respect to uncertainties in the LOS. This involves comparing predictions made using the reference model with fixed LOS at the onset times with predictions made using the reference model and perturbed LOS. The perturbed LOS are created by randomly drawing onset times within the data uncertainty and fixing the

LOS at these perturbed onset times. This process is repeated 2,500 times in order to calculate the prediction covariance matrix associated with uncertainty regarding the LOS.

The final prediction covariance matrix is taken as the sum of the one accounting for uncertainties with respect to the acoustic wave speed profile and the one accounting for uncertainties with respect to the LOS. Figure 8 shows a comparison of the different sources of uncertainty. Prediction uncertainties with respect to the LOS are often very small, whereas prediction uncertainties with respect to acoustic wave speed can often exceed observations uncertainties. Overall, prediction uncertainties account for approximately 77% of the total uncertainty, though this figure can reach 98%.

## 2.3 Bayesian inference of the model parameters

As described above, the model consists of 4 parameters  $\mathbf{m} = [\lambda_{\rm src}, \phi_{\rm src}, h_{\rm iono}, \tau]$ . To infer these parameters from the measured onset times, Bayes' theorem is used:

$$p(\mathbf{m}|\mathbf{d}) \propto p(\mathbf{d}|\mathbf{m})p(\mathbf{m}) \tag{10}$$

where  $\mathbf{d}$  is the data vector. The goal is to obtain the left-hand side of this equation, which is the posterior probability density function (hereafter referred to as the post-PDF). This provides an estimate of the probability of a model based on the observed data. As there are no analytical solutions, the posterior probability density function (post-PDF) is approximated using the product of the prior PDF  $(p(\mathbf{m}))$  and the likelihood  $(p(\mathbf{d}|\mathbf{m}))$ .

The prior PDF reflects the prior beliefs about the model. For instance, it is reasonable to assume that the source location of the CID is not located far from the earthquake's epicentre. A uniform prior is used to restrict the search for the parameters within certain bounds. For the source location of the CID, the latitude and longitude are bounded at  $\pm 2^{\circ}$  away from the earthquake centroid location (3.71°S, 99.32°E). The effective height of the ionosphere is restricted to be between 100 and 600 km. The uniform time shift is restricted between -300 and 300 seconds.

The likelihood quantifies the probability that the observations are compatible with a given model. This is achieved by measuring the discrepancy between the observations and the model's predictions. The most common approach is to use a Gaussian likelihood function, which is equivalent to evaluating the misfit using the  $\mathcal{L}_2$ -norm in an optimisation problem (Minson and Lee, 2014). However, as the data set is expected to contain some outliers, a Laplacian likelihood function is preferred (Minson and Lee, 2014):

$$p(\mathbf{d}|\mathbf{m}) = \prod_{i=1}^{N} \frac{1}{\sqrt{2}\sigma_i} e^{-\frac{\sqrt{2}}{\sigma_i} |\mathbf{d} - \hat{\mathbf{d}}|}$$
(11)

where  $\sigma_i^2$  is the  $i^{th}$  diagonal element of  $\mathbf{C}_{\chi} = \mathbf{C_d} + \mathbf{C_p}$ , which correspond to covariance matrices describing data and prediction uncertainties, respectively.

Although there is a functional form for the post-PDF, analytical solutions do not exist and numerical solutions are computationally infeasible. One approach to overcoming this issue is to efficiently sample the post-PDF and analyse the ensemble in order to characterise the distribution. Post-PDF sampling is performed using a parallel tempering algorithm (Sambridge,

2014). This approach makes it easier to move the sample across low-probability regions, thus ensuring better sampling of the posterior PDF. Parallel sampling of  $p(\mathbf{m}|\mathbf{d})$  is performed using 500 chains of 5000 samples each. The temperature (T) of the chains is drawn between 1 and 1000 using a log-normal distribution. Since only the cold chains (T=1) sample the true post-PDF, 250 of the 500 chains have a temperature set to 1. To remove the influence of the random initial model assigned to each chain, a burn-in phase is used, removing the first 10% of the cold chain samples. In addition, to reduce the correlation between samples, thinning is applied, i.e. that 1 out of every 10 samples is retained for the cold chains. Thus, the post-PDF is obtained from 112,500 samples. Finally, the proposal covariance matrix for drawing a proposed move is tuned so that the acceptance rate of each cold chain is approximately 25%, which is considered optimal for a Metropolis-type algorithm (Gelman et al., 1997).

#### 3 Results

220

230

## 3.1 Source Location of the Coseismic Ionospheric Disturbance

Figure 9a shows the posterior probability density function (PDF) for the source location of the CID. The maximum a posteriori probability (MAP) for the source location is  $3.72^{\circ}S \pm 0.09$  and  $99.32^{\circ}E \pm 0.07$ . This location is approximately 100 km west-southwest (WSW) from the epicentre of the Mentawai earthquake and around 50 km southwest (SW) from the region of maximum uplift. These results are also compared with a tsunami simulation in Yue et al. (2014). This shows that the source of the CID is located at the edge of the region where the tsunami initiates. It also coincides with the centroid location given by the GCMT catalog.

# 3.2 The Effective Height of the Ionosphere

Figure 9b shows the posterior probability density function (PDF) for the effective height of the ionosphere. The maximum a posteriori (MAP) estimate is  $264.5 \text{ km} \pm 6.0 \text{ km}$ . This is consistent with the standard value of 300 km typically used for the effective height of the ionosphere. This result is compared with the electron density profile at the time and location of the earthquake, as provided by the International Reference Ionosphere (Figure 9b). The MAP lies below 350 km, the height at which the electron density is at its maximum. Instead, it lies at the base of the ionosphere, close to the point at which the electron density gradient is at its maximum. This may be due to the fact that observations of the onset times are made for various station-satellite pairs. Thomas et al. (2018) have shown that depending on the LOS, the onset time can occur at heights lower than the maximum ionisation.

#### 3.3 Comparison between observations and predictions

Figure 10 shows a comparison between the observed and predicted onset times. The ensemble of predicted onset times are consistent with the observations. Most of the mismatches are observed for late onset times ( $\gtrsim 1100$  seconds). It should be noted that the predictions have undergone a time correction. The MAP for the constant (i.e., non-station-specific) correction

245

250

term  $(\tau)$  is -126.6  $\pm$  12.6 seconds. This means that the predictions tend to arrive too late, requiring the artificial speed up of the acoustic wave speed profile in order to match the observed onset times. This could suggest that the acoustic wave speed profile is inaccurate. However, the prediction covariance matrix has taken into account uncertainties about the acoustic wave speed profile. Therefore, there may be additional sources of uncertainty not considered in this study that are mitigated by  $\tau$ . For example, the coupling with the geomagnetic field induces a non-constant and anisotropic phase shift (e.g., Rolland et al., 2013; Lee et al., 2018), which is not accounted for in the approach presented here. This phase shift can be significant across an array of many station-satellite pairs.

## 3.4 Uncertainties of the model parameters

Figure 11 shows the correlation matrix between the model parameters, as inferred from the posterior PDF. The threshold between a strong and a weak correlation is set at |0.5|. It shows that the longitude of the source of CID ( $\lambda_{\rm src}$ ) is a fairly independent parameter, as it only correlates strongly with the systematic time shift ( $\tau$ ). All other pairs of parameters correlate relatively strongly. The strongest correlations are observed with respect to  $\tau$ , suggesting that it plays an important role as an adjustment variables. This is further illustrated by the synthetic tests presented in the Supplementary Materials. To quantify the robustness of the results, the information gained from the prior and posterior PDF is calculated using the Kullback–Leibler divergence ( $D_{\rm kl}$ ), as suggested by Gombert et al. (2018). The resulting concentration factors ( $2^{D_{\rm kl}}$ ) range from 9 to 23, indicating that the posterior distributions are significantly narrower than their corresponding priors.

## 4 Discussion

The results presented in the previous section suggest that the source location of the CID does not align precisely with the zone of maximum seafloor uplift, nor with the area from which the tsunami originates, contrary to what is commonly suggested (e.g., Astafyeva and Shults, 2018). The synthetic tests presented in the Supplementary Materials suggest that this is not due to a lack of resolution. Therefore, at first glance, there is no reason to suspect that the source location of the CID is incorrectly characterised.

To further challenge the reliability of the results, an additional inversion is carried out, this time assuming that the acoustic wave propagates through a homogeneous atmosphere. In this case, the acoustic wave speed  $(v_r)$  is part of the inversion process, introducing an additional model parameter. During the inversion process, the speed of the acoustic wave is constrained to lie between 0.1 and 2.5 km/s. Since the speed of the acoustic wave is included in the inversion, there is no need to consider the impact of uncertainty in the acoustic wave speed profile on the predictions. Therefore, only prediction uncertainties associated with the LOS are taken into account, along with observation uncertainties.

Figure 12 shows the results of this inversion. While the effective height of the ionosphere remains relatively similar (around 259 km), the source location of the CID shifts to match the zone of maximum uplift and the area where the tsunami initiates. The inferred acoustic wave speed is  $0.6 \text{ km/s} \pm 0.02$ , consistent with the apparent propagation speed of 0.6–0.8 km/s obtained by Manta et al. (2020). The fact that the inferred source location of the CID differs when using a realistic atmospheric profile

versus a homogeneous atmosphere is inconsistent with the results from the synthetic tests (see Supplementary Materials), which demonstrate that both approaches converge on the same source location of the CID with the same bias relative to the true location. This raises the question of which approach makes the most accurate prediction of the source location of the CID.

One possible explanation for this discrepancy is that the acoustic wave propagates at supersonic speeds. This has been observed following some earthquakes. For example, Astafyeva et al. (2011) suggests that the 2011 M9 Tohoku-oki earthquake in Japan generated shock-acoustic waves that propagated at speeds 30–40% faster than typically observed acoustic wave speeds. A similar observation was made by Sun et al. (2016) after the 2015 M7.8 Nepal earthquake, with an apparent acoustic wave propagation speed of approximately 0.8 km/s. This has been shown to be due to the impulsiveness and amplitude of the source, which induces significant nonlinear effects (Wei et al., 2015), causing the formation of a very fast acoustic wave that reaches the ionosphere at supersonic speeds. It was also highlighted by Inchin et al. (2020b) using numerical simulations. Their study demonstrates that nonlinear effects, particularly in the near-epicentral region, could alter the leading phase front.

Propagation at supersonic speeds is not accounted for with an eikonal solver. However, if the propagation speed is part of the inversion, it is possible that the inferred value will exceed the background acoustic wave speed. To verify this, 3D ray tracing is performed in a realistic atmosphere, and using the eikonal solver, to calculate the apparent propagation speed of acoustic waves along the ray paths. The calculation uses the MAP model presented in Section 3. The average apparent propagation speed across all possible station-satellite pairs is  $0.56 \pm 0.03$  km/s, with values ranging from 0.51 to 0.64 km/s. These values are consistent with the one inferred by the inversion that assumes a homogeneous atmosphere as well as with the values obtained by Manta et al. (2020). Consequently, it appears that a supersonic propagation speed is not necessary to explain the observed onset times.

In the context of earthquake location problems, it has been observed that locations determined using a 1D velocity model in the presence of azimuthal heterogeneities tend to shift hypocentres towards areas of higher velocity (e.g., Havskov et al., 2012; Delouis et al., 2021). As faster velocities are observed in the north-east (NE) direction (see Figure 7), this could explain the difference in source location observed between that obtained with a realistic atmosphere and that obtained with a homogeneous atmosphere. Therefore, using a realistic acoustic wave speed should provide an accurate location of the source of CID because it takes into account the complexity of the propagation paths. In light of this, it is interesting to consider why the source location of the CID does not match with the zone of maximum uplift, and why it is located on the edge of the tsunami initiation zone.

One possible explanation could be related to the fact that the coupling between the ocean and the atmosphere is essentially governed by the vertical velocity of the water column (e.g., Hickey et al., 2009; Inchin et al., 2020a). The unusually long source duration of the Mentawai tsunami-earthquake has likely resulted in slow seafloor uplift, potentially causing the tsunami to initiate slowly and leading to inefficient coupling with the atmosphere. Instead, more efficient coupling may occur at the edge of the tsunami initiation area, where there is a sharp transition from high to low wave height, resulting in rather large vertical velocity (e.g., Saito, 2017).

Another possible explanation is that the point-source assumption is unrealistic for such an extended source and that CID are generated simultaneously from multiple distinct locations. For instance, an extended tsunami wavefront can behave like a series of sources, producing multiple acoustic wavefronts that interact to create complex ionospheric signatures. Therefore,

depending on the observation point, the observed onset time may be related to a different source location of the CID (e.g., Bagiya et al., 2020; Heki, 2024). Consequently, the inferred source location may represent the average source location over the entire area. This could explain why the inferred source location of the CID matches the centroid location reported in the GCMT catalogue.

Alternatively, the detectable CID on the vTEC waveforms may not be related to the maximum seafloor uplift or the initiation of the tsunami. Instead, it may be related to ionospheric disturbances travelling ahead of the tsunami wavefront (Kherani et al., 2015), which would explain why the inferred source location of the CID is at the edge of the tsunami initiation area.

Finally, it has been demonstrated that bathymetry plays a significant role in the generation of acoustic waves (Inchin et al., 2020a). Figure 13 shows the bathymetry profile at the inferred source location. Looking towards the onshore direction, it shows that the tsunami initiates at the transition from low to high seafloor elevation. As several studies have suggested (e.g., Song et al., 2008; Lotto et al., 2017), this particular setup may result in the tsunami having a non-negligible initial horizontal velocity, which significantly affects the initiation stage of the tsunami but is often disregarded. This could, in turn, explain the mismatch between the inferred source location of the CID and the area of maximum uplift and tsunami initiation, by affecting the generation of CID.

Looking towards the offshore direction, Figure 13 also shows that the inferred source location of the CID is at a sharp transition from low to high seafloor elevation. This could increase the vertical velocity of the water column at this location, enabling acoustic waves to be sent into the atmosphere more efficiently here than from the point of tsunami initiation. Another observation from Figure 13 is that the trench position from Bird (2003) used to position the rupture and tsunami models does not quite coincide with the trench position seen in the bathymetry. There is a shift of approximately 15 km. Keeping all other factors the same, simply shifting all the models 15 km towards offshore would place the source location of the CID within the initiation area of the tsunami.

## 5 Conclusions

This study uses measurements of the onset times of Coseismic Ionospheric Disturbances (CIDs) on vTEC waveforms to determine the location of the source. To obtain the most reliable estimate, the forward problem of predicting the travel time of an acoustic wave is solved using a realistic atmospheric profile. The common assumption that CID are generated when acoustic waves reach a fixed altitude has also been relaxed. The source location of the CID is inferred using a Bayesian framework that explicitly incorporates observation errors as well as prediction errors due to uncertainties in the atmospheric profile. This methodology is applied to study the CID generated by the 2010 Mentawai tsunami-earthquake.

Firstly, the results confirm the common assumption that the effective height of the ionosphere is approximately 250 km, placing it below the peak of the F-layer. Secondly, the results show that the inferred source location of the CID does not correspond to the zone of maximum seafloor uplift or the area where the tsunami initiates. A match is found when the atmosphere is assumed to be homogeneous, with an acoustic wave propagation speed of around 0.6 km/s. However, the calculated acoustic wave propagation speed within the realistic atmosphere and over the different ray paths is also 0.6 km/s. Therefore, the

difference in location is not due to the acoustic waves propagating at speeds that cannot be handled when using the realistic atmosphere, as it would be the case for a supersonic acoustic front. Consequently, it can be argued that using a realistic atmosphere provides an accurate location of the CID source because it takes into account the complexity of propagation paths to observation points.

Several hypotheses have been put forward to explain why the source location of the CID does not coincide with the zone of maximum uplift or with the area where the tsunami initiates. One is that the point-source approximation used here is inappropriate for an extended source. Another is that the detected disturbances on the vTEC waveforms could be ATID (ahead-of-tsunami travelling ionospheric disturbances). Second-order effects due to the bathymetry could also have changed the dynamics of the tsunami initiation, which in turn would affect CID generation. Finally, both the earthquake rupture model and the subsequent tsunami model are subject to uncertainty. Based on the discrepancy between the plate boundary model and the trench position inferred from bathymetry, it is possible that the tsunami initiates even further offshore, closer to the CID source location obtained in this study.

In any case, the approach illustrated here provides a valuable compromise between purely observational analyses and more computationally intensive numerical simulations. Future developments should focus on extending the current framework to (i) handle multiple simultaneous CID sources, (ii) account for spatial variability in the effective height of the ionosphere, and (iii) investigate the applicability of this approach to (near-)real-time TEC data analysis streams. These enhancements would help to bridge the gap between research-oriented analyses and operational early warning systems.

## 360 Appendix A: Synthetic tests

# A1 Resolution analysis

It has been shown that the distribution of observation points can introduce bias when determining the source location of the CID (e.g., Zedek et al., 2021; Rolland et al., 2013). As the approach in this study is non-linear, it is not possible to compute a spatial resolution matrix. To study how the distribution of observation points affect the source location of CID, a series of inversions is performed. Artificial sources are placed over a 9 by 9 grid that spans over the entire parameter space explored in the main text (white crosses in Figure A1). For each artificial source, synthetic onset times are generated using the same forward modelling approach as described in the main text (Section 2.2). Note that to generate the artificial onset times, the effective height of the ionosphere is fixed at 250 km and the constant correction term is set to 0 seconds. Then, each of these datasets is used to infer the source location as described in the main text (Section 2.3). The uncertainty level of each dataset is the same as that presented in the main text (Figure 8).

Figure A1 shows the posterior PDF obtained for each inferred source location. First, the results show that the approach used in this study can accurately retrieve the location of the artificial sources when the forward problem is perfectly known. Then, it shows that the highest resolution, i.e., where the posterior PDFs are the smallest, are within a rectangular zone ranging from 98.25°E to 100.25°E and from 3.75°S to 1.75°S. This is the zone where we have the highest density of observation points. But, it also encompasses the epicenter and centroid of the 2010 Mentawai earthquake, the zone of maximum seafloor uplift,

and the region where the tsunami initiates. Therefore, the region with the highest resolution is where the source of CID is most likely to be located. This suggests that the distribution of observation points used in this study is appropriate for accurately determining the source location of the CID generated by the 2010 Mentawai tsunami-earthquake.

# A2 Synthetic test using realistic vTEC waveforms

To thoroughly test the approach outlined in the main text, a more realistic synthetic test has been designed. Synthetic TEC waveforms are computed using the IonoSeis software (Mikesell et al., 2019) with an artificial point source for the CID that is located at the epicentre of the 2010 Mentawai earthquake-tsunami. Acoustic waves are propagated in the same atmospheric model as described in the main text (Figure 6a), except that zonal and meridional winds are not taken into account. Synthetic TEC waveforms are affected by the geomagnetic field, which can induce a non-constant and anisotropic phase shift (e.g., Rolland et al., 2013; Lee et al., 2018). However, as this effect is not taken into account in the approach developed for this study, it seems interesting to keep this effect in order to see its influence on the inferred source location of the CID.

The same steps as those described in the main text are followed: (a) onset times are measured manually without visual cues; (b) onset times are adjusted using the ICCS algorithm, which also provides a measure of the observations uncertainty; (b) prediction uncertainties associated with uncertainties about the windless atmosphere model are calculated, as well as those associated with uncertainties on the lines of sight; and (d) a Bayesian inversion is performed to infer the source location of the CID, the effective height of the ionosphere, and the constant (non-station-specific) correction term.

The results are presented in Figure A2. The first thing to note is that the inferred source location of the CID does not coincide exactly with the location of the virtual source, which is at the epicentre of the Mentawai tsunami-earthquake. It is offset by approximately 30 km in the NNE direction relative to the virtual source. Therefore, even though the posterior PDF gives a low level of uncertainty ( $

415

420

across an array of many station-satellite pairs. As this effect is not accounted for in the approach developed in this study, it could explain the discrepancy between the inferred source location of the CID and the actual location of the virtual source.

These effects may be partially offset by the constant (non-station-specific) correction term which is found to be around -90 seconds. In fact, this value is close to that obtained in the main text when real data is inverted (approximately -127 seconds). This may indicate that the same effects are offset in this synthetic test and during the inversion of real data from the Mentawai earthquake.

It is worth noting that the same posterior PDF for the source location of the CID is obtained when an inversion is performed under the assumption of a homogeneous atmosphere (red line on Figure A2a). As the zonal and meridional winds have been turned off, there are no strong azimuthal heterogeneities from the stratified atmospheric model. Therefore, contrary to what is argued in the main text, an inversion using a homogeneous atmosphere may not be subject to the same biases as an inversion in the presence of strong azimuthal heterogeneities. Another interesting outcome is that the inferred propagation speed of the homogeneous atmosphere is 0.6 km/s. As the location inferred by the two types of inversion is the same, this validates the conclusion that an apparent propagation speed of 0.6 km/s is compatible with the acoustic waves propagation speeds within the realistic atmosphere. This further validates the idea that there is no supersonic acoustic wavefront in the case of the 2010 Mentawai earthquake.

Author contributions. Cedric Twardzik: Conceptualization, Methodology, Formal analysis, Investigation, Writing - Original Draft, Visualization Lucie Rolland: Conceptualization, Resources, Writing - Review and Editing, Supervision, Project administration, Funding acquisition Thomas Dylan Mikesell: Conceptualization, Methodology, Validation, Writing - Review and Editing Edhah Munaibari: Resources, Data Curation, Writing - Review and Editing

Competing interests. The authors declare that they have no conflict of interest.

430 Acknowledgements. This study benefited financial support from the Agence National de la Recherche under the ITEC project (ANR-19-CE04-0003). CT and LR would like to thank Amandine Doudelet for initiating the conversation that led to this study. TDM would like to thank for the support of the Research Council of Norway. EM was funded by the Centre National d'Études Spatiales.

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

**Figure 1.** Schematic of the setup used to measure Coseismic Ionospheric Disturbances (CID) generated by a tsunami, as recorded by ground-based GNSS receivers in the form of Total Electron Content waveforms. Figure adapted from (Kanai et al., 2022).

Figure 2. Map showing the distribution of GNSS receivers (blue triangles), which are used to study the CID generated by the 2010  $M_w$ 7.8 Mentawai earthquake. The yellow star indicates the location of the earthquake epicentre, as provided by the Global Centroid Moment Tensor (GCMT) catalogue. The coloured lines show the spatial extent of the line of sight at an altitude of 300 km for each station-satellite pair, spanning the period from the earthquake origin time, as provided by the GCMT catalogue, to 30 minutes afterwards. The thick black lines show the plate boundaries, as given in Bird (2003).

**Figure 3.** Example of a manual measurement of the onset time of the CID for the station-satellite pair MSAIG-PRN09. The vertical red dashed line shows the earthquake's origin time, and the vertical purple dashed line shows the measured onset time. The blue curve shows the vTEC waveform and the orange and green lines show the first and second time derivatives of the vTEC waveform, respectively.

**Figure 4.** Figure illustrating the update to the onset time made by the ICCS algorithm. This example uses the first time derivative of the vTEC waveforms from satellite PRN29. The waveforms on the left are aligned with respect to the manually measured onset times. The right figure shows the waveforms adjusted by the ICCS algorithm. The red lines at the top show the resulting waveform stack. The time adjustments made by the ICCS algorithm to the onset times for each waveform are shown on the right.

**Figure 5.** (a) Map showing the spatial distribution of the observations, assuming a measurement made at an altitude of 300 km. The dots are colour-coded according to the onset time. The yellow and white stars indicate the locations of the epicentre and centroid, respectively, as provided by the GCMT catalog. Further information about the map can be found in the caption of Figure 2. (b) Spacetime diagram of the observations, assuming a measurement made at an altitude of 300 km. The error bars represent the observation uncertainties obtained from the ICCS algorithm. The distances from the epicentre are calculated from the surface projection of the sub-ionospheric piercing point (SIPP) at the onset time, based on the assumption that the measurement was taken at an altitude of 300 km. The dots are colour-coded according to the azimuth of the SIPP with respect to the earthquake epicentre. The lines illustrate the expected apparent propagation time, assuming a source at the epicentre and propagation within a homogeneous atmosphere.

**Figure 6.** (a) The acoustic wave speed profile (blue line), derived from the MSISE-00 empirical model (Picone et al., 2002) at the time and location of the 2010 M<sub>w</sub>7.8 Mentawai tsunami earthquake. (b) The meridional wind speed profile (blue line) given by horizontal wind model 14 (Drob et al., 2015). Positive values indicate that the wind is blowing northward. (c) The zonal wind speed profile (blue line), also given by horizontal wind model 14 (Drob et al., 2015). Positive values indicate that the wind is blowing eastward. In all sub-figures, the dashed line shows profiles corrected for Earth ellipticity.

**Figure 7.** Theoretical arrival time at an altitude of 300 km for a source at the surface, assuming source-receiver distances ranging from 0 to 900 km. This is calculated using an eikonal solver a realistic, stratified atmosphere. The closer the colour is to yellow, the slower the propagation speed in that direction.

Figure 8. The blue bars show the observation uncertainty of the measured onset times as determined by the ICCS algorithm  $(\sigma_d)$ . The orange bars show the uncertainty in the predicted onset times, accounting for the uncertainty in the acoustic wave speed profile  $(\sigma_{p1})$ . The green bars show the uncertainty in the predicted onset times due to uncertainties regarding the line of sight position  $(\sigma_{p2})$ . The overall uncertainty level used during inversion of the source location of the CID is the sum of the observation and prediction uncertainties, i.e.  $\sigma = \sigma_d + \sigma_{p1} + \sigma_{p2}$ .

**Figure 9.** (a) Map showing the posterior PDF of the source location of the CID, ranging from yellow to dark blue to indicate high to low probabilities, respectively. The yellow and white stars indicate the locations of the epicentre and centroid, respectively, as provided by the GCMT catalog. The background shows the calculated uplift using Okada formulations (Okada, 1992) and using the finite fault model provided by the USGS National Earthquake Information Center (2018). The orange star indicates the maximum uplift. The green dashed line outlines the area with a positive tsunami height as simulated by (Yue et al., 2014). The black dashed line shows the cross-section displayed in Figure 13. Further information about the map can be found in the caption of Figure 2. (b) Electron density profile at the location and time of the 2010 Mentawai earthquake (blue), as calculated using the International Reference Ionosphere (IRI). The red dashed line shows the gradient of the electron density on an arbitrary scale. The posterior PDF shows the effective ionospheric height on an arbitrary scale (orange).

Figure 10. Comparison between the observed onset times (blue bars) and the posterior PDF of the predicted onset times (red bars). Note that a constant (non-station-specific) correction term of about -126.6  $\pm$  12.6 seconds is applied to all the predicted onset times. The waveforms are sorted according to the observed onset times. The red vertical line indicates the origin time of the 2010 Mentawai earthquake.

**Figure 11.** This is a correlation matrix of the model parameters, which were calculated from the samples of the posterior PDF. Red colours indicate positive correlations, while blue colours indicate negative correlations. The values at the centre of each square show the calculated correlation coefficient.

Figure 12. Results inferred when assuming a homogeneous atmosphere with an inferred acoustic propagation speed of  $0.6 \text{ km/s} \pm 0.02$ . As convergence is slower, the burn-in phase is set to 40%. Further information about the figure can be found in the caption of Figure 9.

**Figure 13.** Cross-section of the bathymetry from Becker et al. (2009). The elevation has been exaggerated by a factor of 100. It runs perpendicular to the trench and through the maximum a posteriori (MAP) source location of the CID, as also shown in Figure 9. The red line shows the MAP of the source location of the CID, along with the uncertainty. The black line shows the position of the plate boundary from Bird (2003). The orange line indicates the maximum seafloor uplift computed using the finite fault model of the USGS National Earthquake Information Center (2018) and the Okada formulations (Okada, 1992). The green shaded region shows the spatial extent of tsunami initiation from Yue et al. (2014), as also shown in Figure 9.

**Table 1.** Dataset used in this study for PRN09. The updated onset time is the result of applying the ICCS algorithm (Lou et al., 2013). The onset time error is calculated as the difference between the measured and updated onset times (see main text)

| PRN | Station Name | Measured (s) | Updated (s) | Error (s) |
|-----|--------------|--------------|-------------|-----------|
| 09  | KTET         | 739.4        | 754.4       | 15.0      |
| 09  | MKMK         | 770.3        | 725.3       | 45.0      |
| 09  | MSAI         | 543.5        | 543.5       | 15.0      |
| 09  | NGNG         | 775.5        | 805.5       | 30.0      |
| 09  | PKRT         | 744.5        | 759.5       | 15.0      |
| 09  | PPNJ         | 801.2        | 786.2       | 15.0      |

**Table 2.** Dataset used in this study for PRN12. The updated onset time is the result of applying the ICCS algorithm (Lou et al., 2013). The onset time error is calculated as the difference between the measured and updated onset times (see main text)

| PRN | Station Name | Measured (s) | Updated (s) | Error (s) |
|-----|--------------|--------------|-------------|-----------|
| 12  | BTHL         | 615.7        | 585.7       | 30.0      |
| 12  | BUKT         | 1332.0       | 1347.0      | 15.0      |
| 12  | LEWK         | 610.5        | 610.5       | 15.0      |
| 12  | PBLI         | 662.1        | 677.1       | 15.0      |

**Table 3.** Dataset used in this study for PRN14. The updated onset time is the result of applying the ICCS algorithm (Lou et al., 2013). The onset time error is calculated as the difference between the measured and updated onset times (see main text)

| PRN | Station Name | Measured (s) | Updated (s) | Error (s) |
|-----|--------------|--------------|-------------|-----------|
| 14  | ABGS         | 1033.1       | 1033.1      | 15.0      |
| 14  | BSAT         | 801.2        | 801.2       | 15.0      |
| 14  | KTET         | 842.5        | 842.5       | 15.0      |
| 14  | LNNG         | 662.1        | 677.1       | 15.0      |
| 14  | MKMK         | 662.1        | 692.1       | 30.0      |
| 14  | MSAI         | 971.3        | 911.3       | 60.0      |
| 14  | PARY         | 863.1        | 818.1       | 45.0      |
| 14  | PKRT         | 899.1        | 869.1       | 30.0      |
| 14  | PPNJ         | 899.1        | 854.1       | 45.0      |
| 14  | PSKI         | 749.7        | 794.7       | 45.0      |
| 14  | SLBU         | 801.2        | 816.2       | 15.0      |
| 14  | SMGY         | 780.6        | 810.6       | 30.0      |
| 14  | TRTK         | 703.3        | 748.3       | 45.0      |

**Table 4.** Dataset used in this study for PRN21. The updated onset time is the result of applying the ICCS algorithm (Lou et al., 2013). The onset time error is calculated as the difference between the measured and updated onset times (see main text)

| PRN | Station Name | Measured (s) | Updated (s) | Error (s) |
|-----|--------------|--------------|-------------|-----------|
| 21  | BSAT         | 1095.0       | 1155.0      | 60.0      |
| 21  | NGNG         | 961.0        | 946.0       | 15.0      |
| 21  | PKRT         | 878.5        | 878.5       | 15.0      |
| 21  | PPNJ         | 935.2        | 905.2       | 30.0      |

**Table 5.** Dataset used in this study for PRN25. The updated onset time is the result of applying the ICCS algorithm (Lou et al., 2013). The onset time error is calculated as the difference between the measured and updated onset times (see main text)

| PRN | Station Name | Measured (s) | Updated (s) | Error (s) |
|-----|--------------|--------------|-------------|-----------|
| 25  | BSIM         | 842.5        | 827.5       | 15.0      |
| 25  | LEWK         | 863.1        | 878.1       | 15.0      |
| 25  | PBLI         | 682.7        | 697.7       | 15.0      |
| 25  | UMLH         | 1017.7       | 1002.7      | 15.0      |

**Table 6.** Dataset used in this study for PRN29. The updated onset time is the result of applying the ICCS algorithm (Lou et al., 2013). The onset time error is calculated as the difference between the measured and updated onset times (see main text)

| PRN | Station Name | Measured (s) | Updated (s) | Error (s) |
|-----|--------------|--------------|-------------|-----------|
| 29  | ABGS         | 708.5        | 738.5       | 30.0      |
| 29  | BITI         | 966.1        | 921.1       | 45.0      |
| 29  | BSIM         | 1131.1       | 1131.1      | 15.0      |
| 29  | BTHL         | 878.5        | 893.5       | 15.0      |
| 29  | BUKT         | 682.7        | 652.7       | 30.0      |
| 29  | JMBI         | 775.5        | 730.5       | 45.0      |
| 29  | LEWK         | 1229.0       | 1229.0      | 15.0      |
| 29  | MLKN         | 837.3        | 867.3       | 30.0      |
| 29  | MNNA         | 693.0        | 648.0       | 45.0      |
| 29  | MSAI         | 626.0        | 611.0       | 15.0      |
| 29  | PARY         | 589.9        | 604.9       | 15.0      |
| 29  | PBLI         | 976.4        | 1066.4      | 90.0      |
| 29  | PSKI         | 600.2        | 570.2       | 30.0      |
| 29  | PSMK         | 801.2        | 831.2       | 30.0      |
| 29  | PTLO         | 734.2        | 794.2       | 60.0      |
| 29  | TIKU         | 620.8        | 635.8       | 15.0      |

**Table 7.** Dataset used in this study for PRN30. The updated onset time is the result of applying the ICCS algorithm (Lou et al., 2013). The onset time error is calculated as the difference between the measured and updated onset times (see main text)

| PRN | Station Name | Measured (s) | Updated (s) | Error (s) |
|-----|--------------|--------------|-------------|-----------|
| 30  | BTHL         | 991.9        | 1006.9      | 15.0      |
| 30  | LEWK         | 1280.5       | 1280.5      | 15.0      |
| 30  | PBLI         | 1192.9       | 1357.9      | 165.0     |
| 30  | PSMK         | 1048.6       | 988.6       | 60.0      |

**Figure A1.** Map shows the posterior PDF for each inversion performed using artificial sources located at the red dots. The colours of the posterior PDF range from yellow to dark blue to indicate high to low probabilities, respectively. The dotted square outlines the region where the source location of the CID is best resolved. Further information about the figure can be found in the caption of Figure 2.

Figure A2. (a) Map showing the posterior PDF for the inferred source location of the CID. The white star indicates the location of the artificial CID source. The red line outlines the inferred source location of the CID when assuming a homogeneous atmosphere with an acoustic propagation speed of  $0.60 \text{ km/s} \pm 0.02$ . Further information about the figure can be found in the caption of Figure 2. (b) Posterior PDF shows the inferred effective height of the ionosphere on an arbitrary scale (orange). The black dashed line shows the inferred effective height of the ionosphere under the assumption of a homogeneous atmosphere. The blue line shows the electron density according to the International Reference Ionosphere (IRI), while the red dashed line shows the gradient of the electron density.