# Peer review of "Locating the Coseismic Ionospheric Disturbance from the 2010 Mw7.8 Mentawai, Indonesia, Tsunami-Earthquake"

_EGUsphere, 2025_

## Referee Comment (RC1)

Review of "Locating the Coseismic Ionospheric Disturbance from the 2010 Mw7.8 Mentawai, Indonesia, Tsunami-Earthquake" by C. Twardzik et al., submitted to EGUSphere

Kosuke Heki

In this manuscript, the authors report their attempt to pin-point the acoustic wave source position responsible for the coseismic ionospheric disturbances associated with the 2010 Mentawai earthquake, well known as a typical tsunami earthquake (slow rupture). They develop a sophisticated system to consider various error sources such as velocity profile, neutral winds, effective height of the ionosphere, and use the onset time of the disturbance to invert for the source position, ionospheric altitude, and the bias in arrival time common for all the station-satellite pairs. However, the position they obtained does not match with any of the expected positions, and the authors discuss possible origins of the discrepancy. I understand the authors did a lot of work, and the efforts are worth publishing somewhere. However, the authors would need to do a lot before listing so many explanations for the discrepancy.

The major problem

I would like to point out that the initial design of the whole work is not good. The authors try their new method, never tested before for a case with the well-known acoustic wave source, for one of the most difficult earthquake. This earthquake has a rupture lasting for ~2 minutes and it is ambiguous when the acoustic wave was generated. It is also difficult to pin-point the real source of the acoustic wave. In fact, the excitation source positions and excitation times of acoustic wave and internal gravity wave would not coincide in this earthquake.

So, I would suggest the authors apply the new method to an easy case, e.g. 2011 March 9 foreshock of the Tohoku-oki megathrust (this has a fairly compact source), any of the explosive volcanic eruptions (it is clear that the source coincides with the vent, and the eruption time is known). Then, they could know if the new method really works well, and could attribute the deviation of the estimated source to technical flaws. I do not find good reasons for the authors not to do this. Applying the new method to a difficult case would not be the best way.

I would leave the decision to the handling editor. It would be reasonable to ask the authors to do "Revise & Resubmit". The editor could also let the authors to do "Major Revision" so that they explain why the more reasonable procedure (test it for a simple case) was impossible.

Other problems

1.  Arrival time bias

I think the 126.6 second of common arrival time bias is too large. So far, I tried numerical simulation in several cases (e.g., 2020 Beirut explosion [Kundu et al., 2021 Sci. Rep.], 2023 Turkey earthquake [Bagiya et al., 2023 GRL]) assuming standard atmosphere without considering neutral winds, but the observed and the simulated waveforms did not deviate from each other beyond a minute in the time axis. With the realistic velocity profile and neutral winds, such a large deviation would not occur. This may have something to do with the

observed source position deviation (tau and phi are correlated, according to Fig.11), and need to be clarified (is it conceivable that the assumed acoustic wave generation time was wrong?).

2. Errors in the estimated source positions

I think we can infer the distribution of dark blue small dots around the best-fit position (white star) as estimation uncertainties. Does this correspond to, e.g., 3-sigma error ellipse (almost no possibility beyond the rim of the distribution)? Is it reasonable to assume the 40-50 km deviation from the tsunami maximum position (orange star) completely significant (e.g., possibility < 0.1 percent)?

3. Uniform velocity assumption

The authors try an unrealistic velocity profile (uniform velocity) and find the estimated source geophysically more reasonable. In the real world, the AW ray bends (convex upward) and apparent (average) velocity, assuming straight ray path, would become faster for points farther from the source. I think this would not be negligible in the given geometry of the GNSS station network, and there is no reason to consider this simple assumption provides a better result than the realistic way.